# Training an Open-Vocabulary Monocular
# 3D Object Detection Model *without* 3D Data

**Rui Huang**[1]    **Henry Zheng**[1]    **Yan Wang**[2]    **Zhuofan Xia**[1]    **Marco Pavone**[2,3]    **Gao Huang**[1,4*]

[1]Department of Automation, BNRist, Tsinghua University, China
[2]NVIDIA Research, USA   [3]Stanford University, USA
[4]Beijing Academy of Artificial Intelligence, China
`{hr20, jh-zheng22, xzf23}@mails.tsinghua.edu.cn`
`yanwan@nvidia.com, pavone@stanford.edu, gaohuang@tsinghua.edu.cn`

## Abstract

Open-vocabulary 3D object detection has recently attracted considerable attention due to its broad applications in autonomous driving and robotics, which aims to effectively recognize novel classes in previously unseen domains. However, existing point cloud-based open-vocabulary 3D detection models are limited by their high deployment costs. In this work, we propose a novel open-vocabulary monocular 3D object detection framework, dubbed OVM3D-Det, which trains detectors using only RGB images, making it both cost-effective and scalable to publicly available data. Unlike traditional methods, OVM3D-Det does not require high-precision LiDAR or 3D sensor data for either input or generating 3D bounding boxes. Instead, it employs open-vocabulary 2D models and pseudo-LiDAR to automatically label 3D objects in RGB images, fostering the learning of open-vocabulary monocular 3D detectors. However, training 3D models with labels directly derived from pseudo-LiDAR is inadequate due to imprecise boxes estimated from noisy point clouds and severely occluded objects. To address these issues, we introduce two innovative designs: adaptive pseudo-LiDAR erosion and bounding box refinement with prior knowledge from large language models. These techniques effectively calibrate the 3D labels and enable RGB-only training for 3D detectors. Extensive experiments demonstrate the superiority of OVM3D-Det over baselines in both indoor and outdoor scenarios. The code will be released.

## 1 Introduction

3D object detection has attracted substantial attention in recent years due to its broad applications in fields such as autonomous driving [13, 25, 26, 64, 55, 31], augmented reality [17, 15, 22], embodied AI [41, 7, 73, 61], *etc*. Although many 3D detectors [42, 62, 51, 35, 32, 48, 50] have marked their impressive performance in the course of 3D perception development and consistently renew the state-of-the-art records on the popular benchmarks [14, 10, 52], they often struggle to generalize to other object categories beyond their training data, severely limiting their applications in diverse real-world scenarios. To address this challenge, many previous works [30, 6, 59, 76, 71, 75] have taken steps forward to detect open-vocabulary objects in broader scenarios, *i.e.*, focusing on identifying and localizing the objects that do not belong to the known categories during training.

However, these open-vocabulary 3D detectors rely heavily on point clouds captured from high-precision LiDARs or 3D sensors, which are expensive to deploy. On the contrary, in another line of research, monocular 3D object detectors directly take in single-view images to localize and recognize

---

*Corresponding author.

38th Conference on Neural Information Processing Systems (NeurIPS 2024).

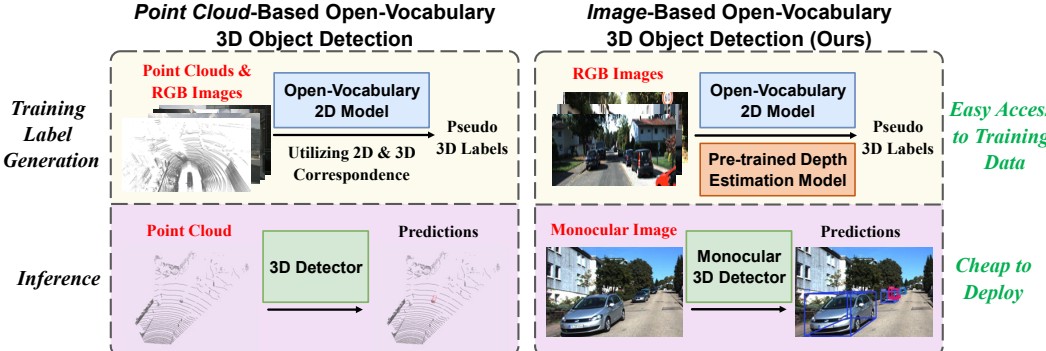

Figure 1: Comparison between point cloud-based and image-based open-vocabulary 3D object detection methods. During training, point cloud-based approaches require corresponding point cloud and image data to derive pseudo labels, while image-based methods can leverage large-scale image data and the most advanced depth estimation models for pseudo-label generation. During inference, point cloud-based methods necessitate expensive LiDAR or other 3D sensors for deployment, whereas image-based approaches only require a camera.

3D bounding boxes of the targets. Without the costly acquisition of point clouds, monocular 3D detectors broaden the horizon of deploying object detection models on more economical platforms [33, 9, 4, 29, 44]. Even though the monocular approaches have liberated 3D object detection from the expensive point clouds during inference, their training procedure still requires images and LiDAR or 3D sensor data pairs in the intensive labeling process, preventing further data scaling. Given the abundance of RGB images online, one might ask: *Could the training of monocular 3D detectors benefit from the readily available RGB images to improve the open-vocabulary perception capability?*

In this paper, we answer this question by proposing a novel open-vocabulary monocular 3D object detection framework named **OVM3D-Det**, which requires only RGB images for training, enables the exploitation of a diverse array of data, and fosters its open-vocabulary potential upon deployment, as shown in Fig. 1. To excavate valuable knowledge and explore the open-set objects from large amounts of images, we harness open-vocabulary 2D models to find the objects that belong to the novel categories. In order to obtain their corresponding 3D bounding boxes, we adopt pseudo-LiDAR [57] assisted with depth estimator to determine the location of the objects in 3D spaces.

Nevertheless, a naive implementation of this framework leads to inaccurate 3D bounding boxes, impairing the performance of trained open-vocabulary 3D detectors (see Tab. 5(a), 7.3% vs. 18.5% AP). We attribute this to the highly noisy pseudo-LiDAR. Although recent advances in depth estimation models [16, 67, 40, 56, 65] showcase impressive zero-shot generalization across various structured environments and cameras with different intrinsics, pseudo-LiDAR can accumulate errors during the auto-labeling process: (1) *the generated point clouds are noisy, making it difficult to distinguish targets from background artifacts* (see Fig. 2); (2) *target objects may be occluded whose actual sizes are hard to estimate, as the point clouds are projected from a single view*.

Facing these challenges, we design two innovative components in our OVM3D-Det framework. In order to mitigate the first issue, we propose an adaptive erosion method to filter artifacts in pseudo-LiDAR and reserve the target objects. This method adaptively removes noise points close to targets according to the object size, thereby improving the precision of 3D bounding boxes. For the second issue, if most points of the objects are missing, accurately estimating their dimensions is challenging. We introduce prior knowledge of objects in each category by prompting large language models such as GPT-4 [1] to determine reasonable bounding boxes and devise a bounding box search technique to optimize the unreasonable box. Finally, the open-vocabulary monocular 3D object detectors are trained to localize with the pseudo bounding boxes and to recognize by aligning with text embeddings.

In summary, the contributions of our paper are threefold: (1) We present the first work targeting image-based open-vocabulary 3D detection tasks across both indoor and outdoor scenarios; (2) We introduce a framework, OVM3D-Det, that automatically labels 3D objects without relying on LiDAR point clouds, thereby expanding the capability to utilize internet-scale data; (3) Our proposed detector, trained on auto-labeled pseudo targets, significantly outperforms the baselines, showcasing the effectiveness of our approach.

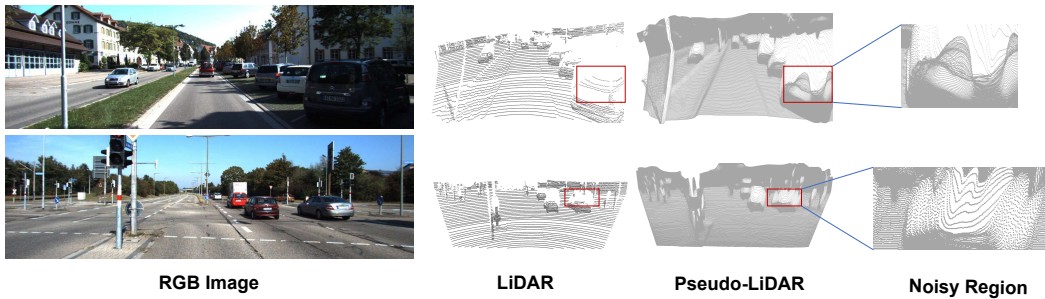

**RGB Image**      **LiDAR**      **Pseudo-LiDAR**      **Noisy Region**

Figure 2: Comparison between LiDAR data and pseudo-LiDAR. Although pseudo-LiDAR is much denser than LiDAR, it is highly noisy (as highlighted in the red boxes), making it inadequate for directly generating accurate 3D bounding boxes.

## 2 Related Works

**Open-Vocabulary 3D Object Detection.** Recently, there has been an increasing interest in 3D open-vocabulary scene understanding. This emerging field takes advantage of the zero-shot recognition capabilities of 2D vision-language models [20, 45], enabling the generalization to objects belonging to new classes [8, 11, 19, 39, 53, 70, 23]. In this context, several works [30, 6, 59, 12, 71, 75] explore open-vocabulary 3D object detection, targeting novel object identification. OV-3DET [30] introduces triplet cross-modal contrastive learning to link image, point cloud, and text modalities, allowing the point cloud detector to leverage vision-language pre-trained models [45]. CODA [6] trains a class-agnostic 3D detector on annotated base categories, gradually discovering novel objects and incorporating them into the training labels. However, previous methods require point clouds as input with high deployment costs, while we focus on open-vocabulary monocular 3D object detection.

**Monocular 3D Object Detection.** Monocular 3D object detection aims to predict 3D bounding boxes and object classes from a single image, which is challenging due to the loss of depth information when converting 3D to 2D. Previous works [33, 9, 4, 29, 74, 44, 63, 37, 72] leverage additional prior information, such as geometric constraints or ground depth, to enhance performance. Pseudo-LiDAR [43, 57, 58, 68] generates pseudo LiDAR points with the off-the-shelf depth estimator and then employs the LiDAR-based 3D detector. Cube R-CNN [3] improves the model's generalization across datasets by utilizing virtual depth, transforming object depth with consistent virtual camera intrinsics. While these methods have yielded impressive results, they all require manually labeled 3D annotations for model training. To reduce the labor-intensive 3D data labeling process, some works [38, 18, 54] have explored weakly-supervised approaches for monocular 3D object detection. Although these approaches eliminate the need for 3D annotations, they still require raw LiDAR data. In [21], to address the lack of 3D annotations, they use an off-the-shelf depth estimator to enhance 3D localization, transferring this information to 3D detectors through self-knowledge distillation. However, with the depth estimation model being pre-trained on the same dataset used for 3D detection, the generalization ability of this method cannot be guaranteed. Different from weakly-supervised methods, we seek to ease the burden of expensive 3D data acquisition, enabling the massive image data and increasing the model's open-vocabulary ability.

**Depth Estimation Model.** Reconstructing depth from monocular RGB image is an ill-posed problem [68, 36]. MiDaS [46] introduces an affine-invariant target to ignore the varying depth scales and shifts across different datasets, estimating the relative depth. Recently, the achievements of visual foundation models [34, 24] trained on large-scale datasets have encouraged the development of various foundational depth estimation models [16, 67, 40, 56, 65]. Depth Anything[65] scales up to a large volume of unlabeled data, significantly expanding data coverage and thereby improving generalization. Metric3D [67] points out that the key to generalization lies in addressing the metric ambiguity of different camera models and proposes a canonical camera space transformation to handle the scale ambiguity problem. Unidepth [40] designs a camera module that generates dense camera representations to guide the depth prediction module, allowing it to zero-shot predict depth for a given image. The tremendous progress of these depth estimation models lays the foundation for generating practical pseudo-LiDAR in our method.

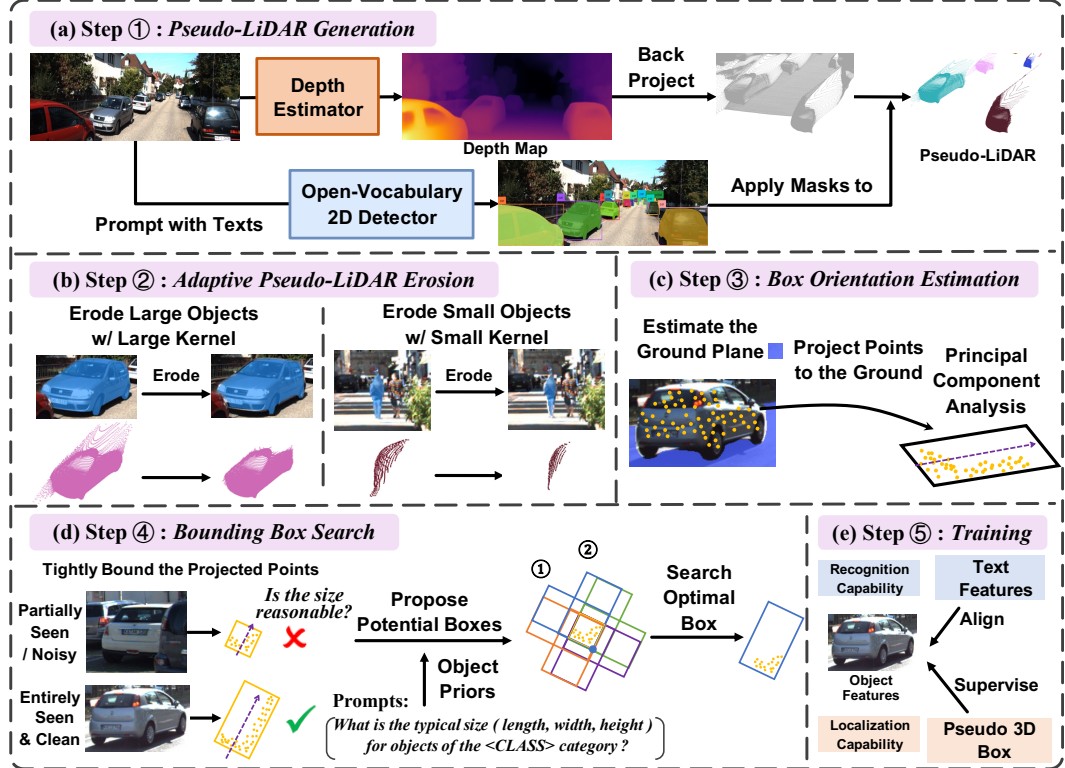

Figure 3: The overall framework of **OVM3D-Det**. Step ①: Generate per-instance pseudo-LiDAR. Step ②: Apply an adaptive erosion process to remove artifacts and noises. Step ③: Estimate the orientation. Step ④: Tightly fit a box and utilize object priors to assess the estimated box; if deemed unreasonable, search for the optimal box. Step ⑤: Train the model with pseudo labels.

## 3 Method

Our goal is to train an open-vocabulary monocular 3D model in an unsupervised manner using image data, without the need for LiDAR point clouds. To achieve this, we propose an automatic labeling framework, OVM3D-Det, that generates 3D labels for RGB images. OVM3D-Det initially generates per-instance depth maps and unprojects them into pseudo-LiDAR point clouds. It subsequently post-processes the instance-level pseudo-LiDAR data to remove artifacts and noises (Sec. 3.1). Then, we fit a 3D bounding box closely to the refined pseudo-LiDAR and employ object priors to evaluate the box; if it is considered unreasonable, we search for the optimal box. (Sec. 3.2). Finally, we train a Cube R-CNN model on the auto-labeled data (Sec. 3.3). The overall framework is illustrated in Fig. 3.

### 3.1 Generate Pseudo-LiDAR

**Pseudo-LiDAR Generation.** For a given image, we begin with using the off-the-shelf Grounded-SAM [47] to detect and segment novel objects. Grounded-SAM integrates Grounding DINO [28] as an open-set object detector, combined with the segment anything model (SAM) [24], enabling simultaneous detection and segmentation of regions within images using arbitrary text inputs. After querying Grounded-SAM with a text list of novel objects of interest, we derive bounding boxes $\{B_k\}_{k=1}^{K}$ and corresponding masks $\{M_k\}_{k=1}^{K}$ for all the objects of interest in the image, where $K$ denotes the number of detected objects. Then we use the pre-trained monocular deep estimation model Unidepth [40] to predict its depth map $D(u, v)$. Note that we directly employ the pre-trained model for zero-shot estimation, and there is no overlap between the datasets used in our experiments and the training datasets of the pre-trained model. This ensures that our method has strong generalizability. Following [57], we can obtain the 3D coordinates $(x, y, z)$ of each pixel $(u, v)$ within the camera's coordinate system:

$$
\begin{aligned}
z &= D(u, v) \\
x &= (u - c_U) \times z / f_U \\
y &= (v - c_V) \times z / f_V
\end{aligned}
\tag{1}
$$

Here, $(c_U, c_V)$ denotes the pixel coordinates corresponding to the camera center, while $f_U$ and $f_V$ represent the horizontal and vertical focal lengths, respectively. For each mask $M_i$, applying eq. (1) to all pixels within the mask yields a 3D point cloud $\left\{\left(x^{(n)}, y^{(n)}, z^{(n)}\right)\right\}_{n=1}^{N_i}$ representing the object, where $N_i$ denotes the total pixel count of the mask $M_i$. The resulting point cloud is referred to as pseudo-LiDAR.

**Adaptive Pseudo-LiDAR Erosion.** We visualize the pseudo-LiDAR and observe that, while it is generally impressive, it still exhibits considerable noise, making it challenging to directly obtain accurate 3D bounding boxes from it. As shown in Fig. 2, compared to LiDAR data, pseudo-LiDAR is denser but also noisier. We identify that the primary source of inaccuracy in pseudo-LiDAR stems from projection errors at the edges of predicted masks. This issue arises because foreground objects and the background, which may have large disparities in depth values, can appear closely adjacent in images, thereby complicating depth estimation at object boundaries. Therefore, we propose employing the image erosion operation to eliminate inaccurate object mask edges while preserving other regions. Image erosion is the morphological operation in digital image processing. Despite its simplicity, the erosion procedure effectively reduces noise, resulting in a cleaner pseudo-LiDAR representation. As shown in Fig. 3(b), after eroding the edges of the car's mask, the pseudo-LiDAR of the car no longer contains noise at the top and front of the vehicle. Please refer to Sec. C for details.

On the other hand, we note that aggressive erosion may completely remove small objects like distant pedestrians, while mild erosion may leave considerable noise on large objects. Hence, we introduce an adaptive erosion module to address this issue. Specifically, this module adjusts the erosion technique based on the Grounding-SAM-generated mask size in the RGB image. Specifically, small objects will undergo less erosion, preserving more of their area, while larger objects will have more of their edges removed, ensuring more accurate pseudo-LiDAR.

### 3.2 Generate Pseudo 3D Bounding Boxes

**Box Orientation Estimation.** Building upon the refined pseudo-LiDAR, we generate 3D bounding boxes as pseudo-labels to equip our monocular 3D object detection model with localization capabilities. Without losing generality, we assume that the 3D bounding boxes of objects in the scene are parallel to the ground. To generate such boxes, we begin by estimating the ground plane equation in the camera coordinate system $C$. The same method as in Sec. 3.1 is used to generate the pseudo-LiDAR for the ground. Specifically, Grounded-SAM [47] is prompted with "ground" for outdoor scenes or "floor" for indoor scenes to obtain their masks. We back-project these masks into 3D space using the estimated depth, and apply the least squares method to fit these points and derive the equation representing the ground. Then, the pseudo-LiDAR is transformed to the new coordinate system $C'$ where the origin aligns with that of the camera coordinate system $C$ but the horizontal plane is parallel to the ground.

As presented in Fig. 3(c), we further project the point cloud onto the ground plane to estimate the orientation of the boxes. Following the method described in [38], we calculate the direction of each pair of points in the pseudo-LiDAR and construct a histogram of these calculated directions. The direction with the highest frequency in the histogram is selected as the object orientation. Additionally, we try Principal Component Analysis (PCA) to extract the main feature components of the pseudo-LiDAR through a linear transformation, assigning the direction with the greatest variance as the orientation of the bounding box. Through experiments, we observe that the simpler and more efficient PCA method can effectively extract the orientation of the objects (see Tab. 5(c)).

**Bounding Box Search.** After estimating the orientation, we enclose the pseudo-LiDAR of each object with a tight 3D bounding box according to the object's direction. These bounding boxes are recorded as coarse pseudo boxes. However, due to the possible occlusion of objects in the images, the generated pseudo-LiDAR may only cover part of the objects. On the other hand, the refined pseudo-LiDAR may still contain noise. As a result, we observe that the enclosed boxes can be

undersized or oversized. Therefore, to generate the proper pseudo labels for training, we further search for the optimal bounding box based on the coarse pseudo box.

We first determine whether the dimensions of a coarse pseudo box are within a reasonable range. Large language models (LLMs), such as GPT-4 [1], are employed to gather data on the typical sizes of objects. For instance, we prompt GPT-4 with "Please provide the (length, width, height) for objects of the <CLASS> category according to their typical sizes in real life."

In our experiments, we find that GPT-4 can offer reliable information regarding the usual sizes of objects within specific categories, including their width, length, and height. When we replace the object sizes predicted by GPT-4 with the sizes statistically derived from the ground truth annotations in the training dataset, the performance is similar (see Tab. 5(d)). The predicted dimensions from GPT-4 are considered as the category priors. If the dimensions of a coarse pseudo box fall within the reasonable range of its corresponding category priors, the coarse pseudo box is considered a valid pseudo label. This range is set as $\tau_1$ to $\tau_2$ times the priors, where $\tau_1$ represents the lower threshold and $\tau_2$ is the upper threshold. Conversely, if it is too small or too large, it undergoes an optimal box search process.

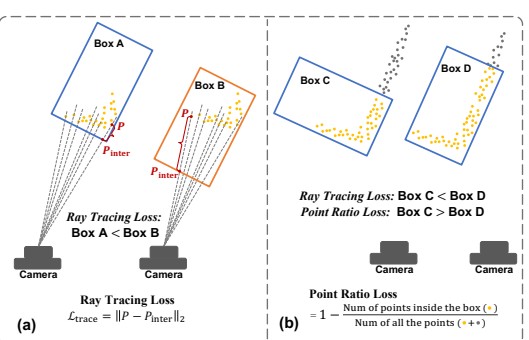

Figure 4: Ray tracing loss and point ratio loss.

Potential boxes are proposed based on the coarse pseudo box. Since the pseudo-LiDAR is constructed from the projection of points captured on the surface of the object, at least one edge of the coarse pseudo box should belong to the optimal pseudo box. Therefore, in the bird's-eye view, for each of the four corners of the coarse pseudo box, we propose two boxes for two possible directions with the class dimension priors (see the blue point and two blue boxes in Fig. 3(d)). That is to say, there are 8 proposal boxes for each coarse pseudo box.

We then select the best-fitting proposal box through a box-searching procedure. Given that monocular RGB cameras only capture the surfaces of objects facing the camera, a valid bounding box should have point clouds concentrated around the edges closer to the camera compared to those farther away. Therefore, we borrow the ray tracing loss from [38] to determine which box is proper. As shown in Fig. 4(a), the loss calculates the distance between each LiDAR point and the nearest intersection point between the proposal box and the camera ray to that point. However, calculating the ray tracing loss alone is not sufficient. Since the calculation of the ray tracing loss only includes points where the camera ray intersects with the box, there can sometimes be shortcuts. As shown in Fig. 4(b), Box C has less ray tracing loss, but it is clearly not what we want. Thus, we propose adding a point ratio loss that considers the ratio of the point cloud enclosed by the box to the overall object point cloud. The overall box-searching loss is the sum of these two losses:

$$\mathcal{L}_{\text{trace}} = \begin{cases} \|P - P_{\text{inter}}\|_2, & \text{if } \text{Ray}_P \text{ intersects with the proposal box,} \\ 0, & \text{otherwise.} \end{cases}$$

$$\mathcal{L}_{\text{point\_ratio}} = 1 - N_{\text{inside}}/N_{\text{all}}, \quad \mathcal{L}_{\text{search}} = \mathcal{L}_{\text{trace}} + \lambda \mathcal{L}_{\text{point\_ratio}},$$
(2)

where $N_{\text{inside}}$ and $N_{\text{all}}$ denote the number of points inside the pseudo box and the total number of points, $\lambda$ is the hyper-parameter that balances the contribution of two losses. For each of the objects, the proposal box with the lowest box-searching loss is considered the pseudo label for training.

## 3.3 Training Detector on Auto-Generated Pseudo Labels

We adopt Cube R-CNN [3] as our basic 3D monocular detector, and replace its classification branch with a text-alignment head that aligns the output object features with text embeddings. The aligning loss $\mathcal{L}_{\text{aligning}} = \mathcal{L}_{\text{CE}}(\mathbf{c}_i, y_i)$, where $\mathbf{c}_i = f_i \cdot \mathbf{t}$ represents the dot product of the object feature $f_i$ and text embeddings $\mathbf{t}$, and $y_i$ denotes the pseudo label predicted by Grounded-SAM. The training objective is defined as: $\mathcal{L}_{\text{train}} = \mathcal{L}_{\text{localization}} + \mathcal{L}_{\text{aligning}}$.

# 4 Experiments

We first evaluate the proposed OVM3D-Det model on outdoor datasets, KITTI [14] and nuScenes [5], as well as indoor datasets, SUN RGB-D [52] and ARKitScenes [2] (Sec. 4.2). We then provide thorough ablation studies and detailed analysis to uncover the factors contributing to the effectiveness of OVM3D-Det (Sec. 4.3). Finally, we show qualitative results of our OVM3D-Det model (Sec. 4.4).

## 4.1 Experiment Setup

**Datasets.** We evaluate OVM3D-Det on the KITTI [14] and nuScenes [5] datasets for outdoor settings, and on the SUN RGB-D [52] and ARKitScenes [2] datasets for indoor settings. Note that neither of these datasets appeared in the training data of Unidepth [40], so there is **no data leakage**. KITTI [14] has 7,481 images for training and 7,518 images for testing. Since the official test set is unavailable, we follow [3] to resplit the training set into 3,321 training images, 391 validation images, and 3,769 test images. For nuScenes [5], we use 26,215 images for training, 1,915 images for validation, and 6,019 images for testing. SUN RGB-D [52] consists of a total of 10k samples, each annotated with oriented 3D bounding boxes, in which 4,929 samples are used for training, 356 samples for validation, and 5,050 samples for test following [3]. ARKitScenes [2] includes 48,046 images for training, 5,268 images for validation, and 7,610 images for testing.

**Metrics.** We report 3D average-precision ($AP_{3D}$) following Cube R-CNN [3] for fair comparison. Predictions are matched to ground truth by calculating the intersection-over-union ($IoU_{3D}$) of 3D bounding boxes. We report $mAP_{3D}$ across all categories over the $IoU_{3D}$ thresholds from 0.05 to 0.50 at a 0.05 step. We split the dataset into base and novel categories to evaluate open-vocabulary performance. In KITTI, there are 2 base classes (car, pedestrian) and 3 novel classes (cyclist, van, truck); nuScenes has 3 base and 6 novel classes; SUN RGB-D includes 20 base and 18 novel classes; and ARKitScenes has 8 base and 6 novel classes. Please refer to Sec. A for more details. We would like to note that we do not need to split the datasets into base and novel splits to train our model. We introduce this base/novel setting only to compare our method with the proposed baselines due to the lack of a suitable baseline before.

**Implementation Details.** We prompt the off-the-shelf Grounded-SAM [47] to detect and segment objects in new categories. For the depth estimation, we adopt the state-of-state-art monocular depth estimation model, Unidepth [40]. Both models are used directly for inference without any fine-tuning. We train on the monocular 3D object detection model Cube R-CNN [3] with the auto-generated pseudo labels, modifying the classification branch to output a 768-dimensional feature and following most of its optimization hyper-parameters. See Sec. E for more details.

## 4.2 Main Results

**Baselines.** To the best of our knowledge, we are the first to explore open vocabulary monocular 3D object detection models. Therefore, to better evaluate the proposed task, we introduce the following baselines: 1) We train the Cube R-CNN [3] with all dataset annotations, including both base and novel classes, to establish the oracle baseline. 2) We develop another baseline to enable Cube R-CNN with open vocabulary capabilities by utilizing Grounding DINO. Specifically, we first use the base class annotations of the training set to train a class-agnostic Cube R-CNN detection model. This enables it to localize the instances from the novel classes during testing, without being limited to a fixed set of categories. During testing, we use Cube R-CNN to predict 3D bounding box proposals and use Grounding DINO to obtain 2D detection results from the images. We then back-project the 3D bounding boxes onto the image frames and match them with the predictions of Grounding DINO. The matched 2D detection boxes from Grounding DINO are then used to assign categories to the corresponding 3D bounding boxes. 3) Although our OVM3D-Det model does not require any annotations for training, we provide a baseline for comparison that is trained using ground-truth annotations for base classes and pseudo labels for novel classes, denoted by OVM3D-Det*.

**Results.** We report the open-vocabulary monocular 3D object detection results on outdoor and indoor datasets in Tab. 1 and Tab. 2, respectively. It can be observed that the unsupervised OVM3D-Det model significantly outperforms the baseline on novel categories. Specifically, it achieves a 6.7%

Table 1: Open-vocabulary monocular 3D object detection results on KITTI and nuScenes. To compare with the baseline, we also present the OVM3D-Det results trained using ground-truth annotations for base classes and pseudo labels for novel classes, denoted by ∗.

| Method | GT Labels | | KITTI | | nuScenes | |
|---|---|---|---|---|---|---|
| | Base | Novel | $AP_B$ | $AP_N$ | $AP_B$ | $AP_N$ |
| Oracle Cube R-CNN [3] | Yes | Yes | 46.3 | 21.6 | 30.7 | 27.1 |
| Cube R-CNN [3] + Grounding DINO [28] | Yes | No | 46.8 | 1.5 | 25.6 | 2.1 |
| OVM3D-Det∗ | Yes | No | 46.4 | 9.3 (+7.8) | 32.2 | 11.7 (+9.6) |
| OVM3D-Det | No | No | 33.9 | 8.2 (+6.7) | 13.3 | 11.8 (+9.7) |

Table 2: Open-vocabulary monocular 3D object detection results on SUN RGB-D and ARKitScenes. To compare with the baseline, we also present the OVM3D-Det results trained using ground-truth annotations for base classes and pseudo labels for novel classes, denoted by ∗.

| Method | GT Labels | | SUN RGB-D | | ARKitScenes | |
|---|---|---|---|---|---|---|
| | Base | Novel | $AP_B$ | $AP_N$ | $AP_B$ | $AP_N$ |
| Oracle Cube R-CNN [3] | Yes | Yes | 15.7 | 14.4 | 39.1 | 36.8 |
| Cube R-CNN [3] + Grounding DINO [28] | Yes | No | 11.0 | 1.5 | 22.2 | 2.9 |
| OVM3D-Det∗ | Yes | No | 16.4 | 9.7 (+8.2) | 41.5 | 21.3 (+18.4) |
| OVM3D-Det | No | No | 11.4 | 10.0 (+8.5) | 20.3 | 19.7 (+16.8) |

Table 3: Comparison with point cloud-based open-vocabulary 3D object detection methods on SUN RGB-D.

| Model | AP |
|---|---|
| Oracle Cube R-CNN [3] | 15.1 |
| OV-3DET [30] (trained with point cloud) | 3.4 |
| OV-3DET [30] (trained with pseudo-LiDAR) | 7.1 |
| CODA [6] | 7.7 |
| OVM3D-Det | 10.8 |

Table 4: Comparison with point cloud-based open-vocabulary 3D object detection methods on KITTI.

| Model | AP |
|---|---|
| Oracle Cube R-CNN [3] | 31.4 |
| OV-3DET [30] | 1.3 |
| OVM3D-Det | 18.5 |

improvement in average precision (AP) on the KITTI dataset, 9.7% AP on the nuScenes dataset, 8.5% AP on the SUN RGB-D dataset, and 16.8% AP on the ARKitScenes dataset. Besides, the OVM3D-Det model trained on the ground truth annotations of the base class and the pseudo labels of the novel class also surpasses the baseline by a large margin on the novel category. This indicates that our framework does indeed generate accurate pseudo-labels for various classes, thereby enhancing the model's performance in the novel category.

In Fig. 6, we show the model performance varying with the amount of training data on KITTI. As the training data increases, we observe a continuous improvement in performance. This highlights the strong potential of our model, suggesting that it can be trained to achieve more generalized open-vocabulary capabilities when using more readily available image data.

**Comparison with Point Cloud-Based Methods.** To compare our method with state-of-the-art point cloud-based open-vocabulary 3D detection approaches, we first simply evaluate existing open vocabulary detector OV-3DET [30] that is trained and tested on point cloud data using pseudo-LiDAR as input. Specifically, we only replace the point cloud input of OV-3DET with the pseudo-LiDAR which is unprojected from the RGB images and estimated depth maps. Other parts of the model in OV-3DET are kept untouched for fair comparison. As shown in Tab. 3, directly changing the input data format leads to poor performance with only 3.4% AP. This phenomenon is also observed and discussed in previous works [36, 66, 60], which we attribute to the distribution shifts between real point clouds and pseudo-LiDARs.

After being trained with pseudo-LiDAR, OV-3DET improves from 3.4% to 7.1% AP, marking a 2x improvement compared to the original model. CODA [6] shows a slight improvement compared

Table 5: Ablation studies. Default settings are marked in gray.

(a) Compare w/ Naive Framework

| Framework | AP |
| --- | --- |
| naive | 7.3 |
| ours | **18.5** |

(b) Mask Erosion

| Erosion | AP |
| --- | --- |
| fixed | 16.4 |
| adaptive | **18.5** |

(c) Orientation Estimation

| Orientation | AP |
| --- | --- |
| histogram | 16.7 |
| PCA | **18.5** |

(d) Box Prior

| Dimension prior | AP |
| --- | --- |
| w/o | 14.0 |
| dataset statistics | 18.4 |
| LLM | **18.5** |

(e) Box Search Loss

| Ray tracing | Point ratio | AP |
| --- | --- | --- |
| ✓ | | 12.2 |
| | ✓ | 17.7 |
| ✓ | ✓ | **18.5** |

(f) Lower Threshold

| $\tau_1$ | AP |
| --- | --- |
| 0.6 | 17.6 |
| 0.7 | 18.0 |
| 0.8 | **18.5** |
| 0.9 | 18.4 |

(g) Upper Threshold

| $\tau_2$ | AP |
| --- | --- |
| 1.1 | 18.5 |
| 1.2 | **18.5** |
| 1.3 | 18.2 |
| 1.4 | 18.1 |

to OV-3DET. Nonetheless, the 10.8% AP of our method still demonstrates our superiority, since our approach focuses on adapting to the noisy nature of pseudo-LiDAR, whereas previous open-vocabulary 3D detectors are mainly designed for real point clouds to generate pseudo labels. The key designs that distinguish our method from other baselines are the adaptive pseudo-LiDAR erosion and bounding box refinement techniques, incorporating the prior knowledge of language models. Our method can generate pseudo labels more effectively for pseudo-LiDAR than other baselines.

For outdoor 3D detection, we also try to establish another baseline since OV-3DET [30] only includes results on indoor data. We attempt to train OV-3DET on the KITTI dataset with pseudo-LiDAR. As presented in Tab. 4, simply training on pseudo-LiDAR points causes a rather weak performance of only 1.3% AP. This result reflects the nature of outdoor data that the vast range of spatial scale could lead to intolerable pseudo-label errors from tiny noise. Therefore, it is challenging to generate reasonable and reliable pseudo boxes due to the inaccurate object edges of the raw pseudo-LiDAR.

## 4.3 Ablation Studies

We conduct ablation studies on the KITTI dataset. The results are shown in Tab. 5. We report the mean AP performance across all the classes for each model.

**Comparison with Naive Framework.** If we remove the refinement applied to pseudo-LiDAR and bounding box (i.e., directly utilizing raw pseudo-LiDAR and tightly bounding it to obtain the box), there is a significant decrease of 11.2% AP in model performance. This result shows the effectiveness of our framework in addressing the noisy pseudo-LiDAR, where the refinement of both the pseudo-LiDAR data and the estimated bounding boxes proves to be pivotal.

**Mask Erosion.** We replace the adaptive erosion process with a fixed kernel erosion, resulting in a decline of 2.1% AP. It demonstrates the effectiveness of our proposed adaptive erosion module, indicating that a standard fixed kernel may not be suitable for all cases, as it could overly erode small objects or be inadequate for larger objects with substantial noise.

**Orientation Estimation.** Here, we compare two methods for determining the orientation of boxes: using Principal Component Analysis (PCA) or calculating the distribution of the directions of each pair of points in the pseudo-LiDAR. We find that the simple and efficient PCA method performs better, exceeding 1.8% AP.

**Dimension Priors.** The dimension priors of objects are an essential component of our method. They not only assist in identifying unreasonable predictions of bounding boxes but also aid in correcting the sizes of the bounding boxes. As shown in Tab. 5(d), the absence of these priors leads to a significant decrease of 4.5% AP. Notably, when we utilize priors derived from a large language model or dataset statistics, the results are similar. In Sec. B, we provide a thorough comparison of LLM-generated priors and shape priors obtained from dataset statistics across all datasets. This suggests that the large language model possesses a strong commonsense understanding, and further underscores the

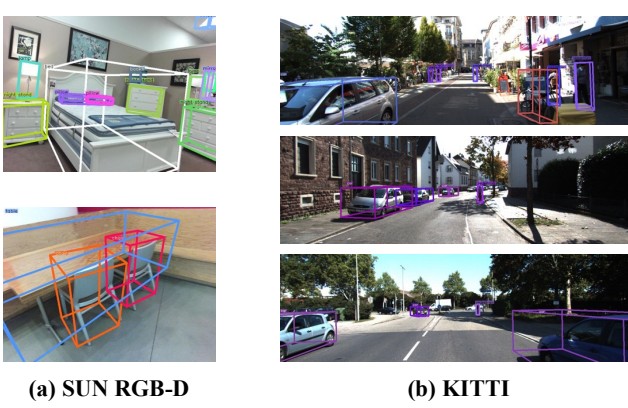

| (a) SUN RGB-D | (b) KITTI |

Figure 5: Qualitative results on SUN RGB-D and KITTI.

Figure 6: Effect of training data. As the volume of training data grows, we consistently see performance improvements.

potential of our approach to be transferred to a wide range of data. We also test the sensitivity of the lower threshold $\tau_1$ and the upper threshold $\tau_2$ used for determining the reasonability of a box. As shown in Tab. 5(f) and (g), they are stable within a wide range.

**Box Search Loss.** We further explore the impact of different loss functions on evaluating the bounding box proposals. The results are presented in Tab. 5(e). Using ray tracing loss alone results in inadequate performance, yielding only a 12.2% AP. However, adding the point ratio loss significantly boosts performance to 18.5% AP.

**Self-Training.** Additionally, we can refine the model using a self-training approach, which involves using the results from the previously well-trained model as pseudo labels in the subsequent training process. As shown in the Tab. 6, self-training can further enhance the quality of the initially generated pseudo boxes, improving detection performance for objects at various distances.

Table 6: Self-training can further improve the quality of the initially generated pseudo boxes.

| Model | AP | AP-Near | AP-Middle | AP-Far |
|---|---|---|---|---|
| OVM3D-Det | 18.5 | 35.7 | 22.0 | 4.5 |
| OVM3D-Det + self-training | 19.9 | 37.6 | 24.6 | 6.0 |

### 4.4 Qualitative Results

The qualitative results are presented in Fig. 5. We prompt the OVM3D-Det model with text queries of the desired categories, obtaining corresponding detection boxes. Results show its capability to locate and recognize objects of diverse types and sizes in both indoor and outdoor scenarios.

## 5 Conclusion

In this paper, we introduced OVM3D-Det, the first framework targeting image-based 3D open-vocabulary detection. OVM3D-Det automatically labels open-vocabulary 3D bounding boxes using only RGB images, exploring the great potential of leveraging omnipresent image data from the internet. To address the noisy point clouds and occluded target objects in predicted pseudo-LiDAR, we devise two innovative solutions: adaptive pseudo-LiDAR erosion and bounding box search using world knowledge. Our open-vocabulary detector trained under OVM3D-Det outperforms the baselines by at least +6.7% AP and up to +16.8% AP. In a word, we propose a simple, yet surprisingly effective baseline method for image-based 3D open-vocabulary detection and we hope it could be a new study point for future research.

**Acknowledgement.** This work is supported in part by the National Key R&D Program of China under grants 2022ZD0114903.

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

# Appendix

## A  Category Splits

The category splits for all datasets are shown in Tab. 7.

Table 7: Category splits for open-vocabulary 3D detection.

| Dataset | Base classes | Novel classes |
|---|---|---|
| KITTI | car, pedestrian | cyclist, van, truck |
| nuScenes | car, pedestrian, trailer | truck, traffic cone, barrier, motorcycle, bicycle, bus |
| SUN RGB-D | bed, chair, bathtub, sink, table, laptop, desk, bookcase, cup, box, oven, machine, bottle, shoes, window, blinds, floor mat, picture, television, bin | toilet, sofa, refrigerator, bicycle, lamp, pillow, nightstand, counter, stove, clothes, shelves, door, curtain, books, cabinet, mirror, stationery, towel |
| ARKitScenes | bed, bathtub, table, television, chair, cabinet, toilet, refrigerator | oven, stove, sofa, machine, shelves, sink |

## B  Dimension Priors

Tab. 8 to 11 provide a comprehensive comparison between LLM-generated priors and shape priors derived from dataset statistics across all datasets. For clarity, we keep one decimal place. As we use shape priors solely to filter out pseudo boxes that may be unreliable due to noise or occlusion and to refine them, it suffices as long as the shape priors fall within a reasonable range.

Table 8: LLM-generated priors and real priors of KITTI dataset.

| Class | Stat. | LLM | Class | Stat. | LLM |
|---|---|---|---|---|---|
| car | [1.6, 1.5, 3.9] | [1.8, 1.5, 4.5] | truck | [2.6, 3.4, 9.3] | [2.5, 3.5, 10.0] |
| van | [1.9, 2.2, 5.1] | [2.0, 2.0, 5.0] | cyclist | [0.6, 1.7, 1.8] | [0.6, 1.7, 1.5] |
| pedestrian | [0.6, 1.8, 0.8] | [0.5, 1.7, 0.8] | | | |

Table 9: LLM-generated priors and real priors of nuScenes dataset.

| Class | Stat. | LLM | Class | Stat. | LLM |
|---|---|---|---|---|---|
| car | [1.9, 1.8, 4.7] | [1.8, 1.5, 4.5] | bicycle | [0.6, 1.4, 1.7] | [0.6, 1.2, 1.8] |
| pedestrian | [0.7, 1.8, 0.7] | [0.5, 1.7, 0.8] | traffic cone | [0.4, 1.1, 0.4] | [0.3, 0.7, 0.3] |
| barrier | [0.5, 1.0, 2.5] | [0.5, 2.0, 2.0] | motorcycle | [0.8, 1.5, 2.1] | [0.8, 1.2, 2.0] |
| truck | [2.6, 3.0, 7.1] | [2.5, 3.5, 8.0] | bus | [3.0, 3.6, 11.4] | [2.8, 3.5, 11.0] |
| trailer | [3.0, 3.8, 11.6] | [2.8, 3.3, 11.0] | | | |

## C  Image Erosion Process

We propose to employ the image erosion operation to eliminate inaccurate object mask edges, thereby obtaining accurate pseudo-LiDAR after unprojection. Image erosion is a fundamental operation in digital image processing that involves shrinking or wearing away the boundaries of foreground objects in a binary image. As shown in Fig. 7, $M_i$ is the binary mask of one object we interested, where each pixel $M_i(x, y)$ is either 0 (background) or 1 (foreground). Let $B(u, v)$ represent the elements of the structuring element $B$, which is a small binary matrix of size $m \times n$. Erosion of the mask $M_i$ by the structuring element $B$ is denoted as $M_i \ominus B$.

Table 10: LLM-generated priors and real priors of ARKitScenes dataset.

| Class | Stat. | LLM | Class | Stat. | LLM |
|---|---|---|---|---|---|
| refrigerator | [0.7, 1.7, 0.7] | [0.8, 1.5, 0.8] | chair | [0.5, 0.8, 0.5] | [0.5, 1.0, 0.5] |
| oven | [0.6, 0.7, 0.6] | [0.6, 0.8, 0.8] | machine | [0.6, 0.9, 0.6] | [0.8, 1.0, 1.0] |
| stove | [0.7, 0.2, 0.6] | [0.6, 0.8, 0.8] | shelves | [0.4, 1.2, 0.8] | [0.3, 1.5, 1.5] |
| sink | [0.5, 0.2, 0.4] | [0.5, 0.2, 0.8] | cabinet | [0.5, 0.9, 0.9] | [0.5, 1.5, 1.0] |
| bathtub | [0.8, 0.6, 1.7] | [0.8, 0.5, 1.5] | toilet | [0.4, 0.7, 0.6] | [0.4, 0.8, 0.5] |
| table | [0.6, 0.6, 1.0] | [0.8, 0.8, 1.5] | bed | [1.6, 0.6, 2.1] | [1.5, 0.5, 2.0] |
| sofa | [1.0, 0.8, 1.4] | [1.0, 1.0, 2.0] | television | [0.9, 0.6, 0.1] | [1.0, 0.5, 0.1] |

Table 11: LLM-generated priors and real priors of SUN RGB-D dataset.

| Class | Stat. | LLM | Class | Stat. | LLM |
|---|---|---|---|---|---|
| bin | [0.4, 0.6, 0.4] | [0.5, 0.5, 0.5] | stove | [0.6, 0.6, 0.8] | [0.6, 0.8, 0.8] |
| box | [0.4, 0.4, 0.4] | [0.5, 0.5, 0.5] | table | [0.8, 0.7, 1.3] | [0.8, 0.8, 1.5] |
| cup | [0.1, 0.2, 0.1] | [0.1, 0.1, 0.1] | pillow | [0.4, 0.3, 0.6] | [0.3, 0.3, 0.5] |
| desk | [0.8, 0.8, 1.4] | [0.6, 0.8, 1.2] | toilet | [0.7, 0.8, 0.5] | [0.4, 0.8, 0.5] |
| bed | [1.6, 1.1, 2.0] | [1.5, 0.5, 2.0] | bathtub | [0.8, 0.5, 1.4] | [0.8, 0.5, 1.5] |
| door | [0.3, 1.8, 0.7] | [0.1, 2.0, 1.0] | towel | [0.2, 0.4, 0.4] | [0.2, 0.1, 0.3] |
| lamp | [0.4, 0.7, 0.4] | [0.3, 0.6, 0.3] | blinds | [0.2, 1.2, 1.2] | [0.1, 1.0, 1.5] |
| oven | [0.6, 0.8, 0.6] | [0.6, 0.8, 0.8] | bottle | [0.2, 0.3, 0.2] | [0.1, 0.3, 0.1] |
| sink | [0.5, 0.4, 0.6] | [0.5, 0.2, 0.8] | laptop | [0.4, 0.2, 0.4] | [0.3, 0.1, 0.4] |
| sofa | [1.0, 0.8, 1.9] | [1.0, 1.0, 2.0] | mirror | [0.2, 1.1, 0.8] | [0.1, 1.0, 0.5] |
| books | [0.3, 0.2, 0.3] | [0.2, 0.1, 0.3] | window | [0.2, 1.1, 1.7] | [0.1, 1.0, 1.5] |
| chair | [0.6, 0.8, 0.6] | [0.5, 1.0, 0.5] | bicycle | [1.0, 1.1, 1.0] | [0.5, 1.0, 1.5] |
| refrigerator | [0.7, 1.5, 0.8] | [0.8, 1.5, 0.8] | picture | [0.1, 0.5, 0.5] | [0.1, 0.5, 0.5] |
| shelves | [0.4, 1.0, 1.3] | [0.3, 1.5, 1.5] | bookcase | [0.4, 1.5, 1.4] | [0.3, 2.0, 1.0] |
| clothes | [0.4, 0.5, 0.5] | [0.5, 1.0, 0.5] | counter | [0.9, 0.9, 2.1] | [0.6, 1.0, 1.5] |
| curtain | [0.3, 1.7, 1.1] | [0.1, 1.5, 1.0] | floor mat | [0.6, 0.1, 0.9] | [1.0, 0.1, 1.5] |
| stationery | [0.3, 0.2, 0.3] | [0.3, 0.3, 0.3] | television | [0.3, 0.6, 0.8] | [0.1, 0.5, 1.0] |
| night stand | [0.5, 0.7, 0.6] | [0.4, 0.5, 0.5] | machine | [0.6, 0.7, 0.6] | [0.8, 1.0, 1.0] |
| shoes | [0.3, 0.1, 0.3] | [0.2, 0.1, 0.3] | cabinet | [0.5, 1.1, 1.2] | [0.5, 1.5, 1.0] |

$$(M_i \ominus B)(x,y) = \begin{cases} 1 & \text{if } M_i(x+u, y+v) = 1 \text{ for all } (u,v) \text{ such that } B(u,v) = 1, \\ 0 & \text{otherwise.} \end{cases}$$

In our experiment, we employ a 3x3 structuring element $B$ where all elements are set to 1. The erosion operation can be applied repeatedly. The intensity of erosion can be controlled by adjusting the number of erosion iterations. We note that aggressive erosion may completely remove small objects like distant pedestrians, while mild erosion may leave considerable noise on large objects. Therefore, for masks with a smaller area, we use a smaller number of iterations; conversely, for masks with a larger area, we use a larger number of iterations, ensuring more accurate pseudo-LiDAR.

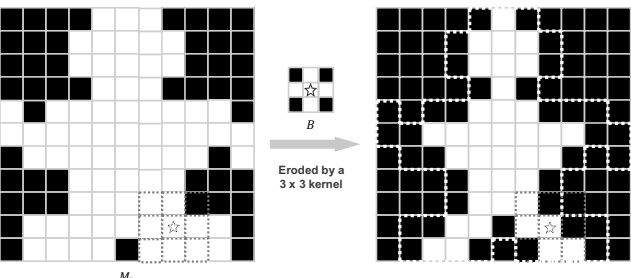

Figure 7: Diagram of the erosion process of $M_i$. To perform the erosion operation $M_i \ominus B$, first place the structuring element $B$ over each pixel of $M_i$. If every foreground pixel of $B$ aligns with a foreground pixel of $M_i$, then the central pixel of $B$ in $M_i$ will retain the value of 1. If not, that central pixel will be set to 0.

## D    Performance Varies with Depth Estimation Quality

Besides Unidepth, we evaluate alternative depth estimation models, such as Metric3D [67]. As shown in Tab. 12, we find that the more accurate the depth estimation model we use, the better the performance of the trained open-vocabulary monocular 3D detection model, suggesting promising potential as depth estimation models advance.

Table 12: Depth Estimation and 3D Detection Performance on KITTI dataset.

| KITTI | Depth Estimation (AbsRel ↓) | 3D Detection (AP ↑) |
|---|---|---|
| Metric3D | 5.33 | 17.0 |
| Unidepth | 4.21 | 18.5 |

## E    Implementation Details

When searching for the optimal box, we set $\lambda$ to 5 for indoor scenes and 10 for outdoor scenes to balance the ray tracing loss and point ratio loss. For the adaptive erosion module, for outdoor scenes, if the maximum width of the instance mask exceeds 10 pixels, image erosion is performed for 4 iterations; otherwise, 2 iterations. For indoor scenes, the values are 12 and 2 iterations, respectively. As we aim to evaluate the benchmarks, we utilize the label lists from the datasets to query the open-vocabulary 2D model for generating pseudo labels of the classes of interest. Following Cube R-CNN, we utilize the DLA34 backbone [69]-FPN [27], which is pretrained on ImageNet [49]. We employ the SGD optimizer, with the learning rate decaying by a factor of 10 at 60% and 80% of the training process. During training, we apply random data augmentation techniques such as horizontal flipping and scaling within the range of [0.50, 1.25]. The model is trained for 29,000 iterations with a batch size of 32 and a learning rate of 0.02 on both SUN RGB-D [52] and ARKitScenes [2], a batch size of 16 and a learning rate of 0.01 on KITTI [14], and a batch size of 32 and a learning rate of 0.01 on nuScenes [5]. We train on the KITTI dataset for ∼ 8 hours with 2 A100 GPUs, on nuScenes for ∼ 6 hours with 4 A100 GPUs, on SUN RGB-D for ∼ 12 hours with 2 A100 GPUs, and on ARKitScenes for ∼ 20 hours with 2 A100 GPUs.

## F    Failure Case

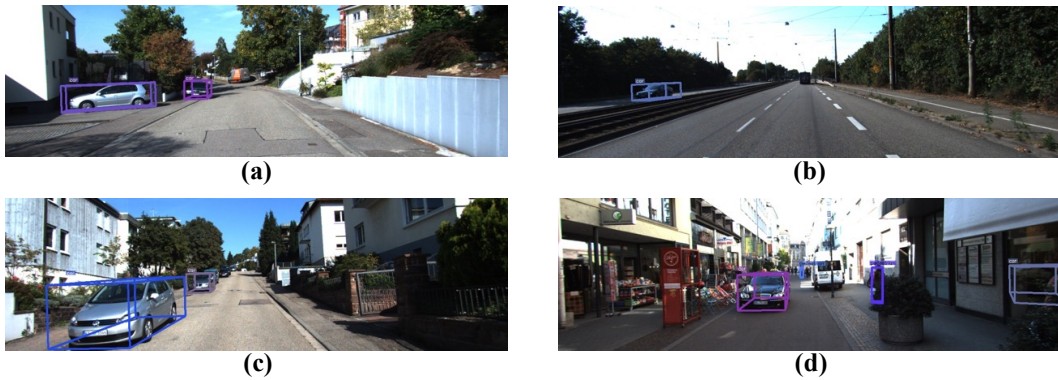

| (a) | (b) |
| (c) | (d) |

Figure 8: Failure case.

We present some failure cases of OVM3D-Det in Fig. 8. (a), (b) and (c) are instances of distant objects being missed by detection. In (d), the reflection of a car on the window is mistakenly detected as a vehicle. As our framework relies on foundational depth estimation models for generating pseudo-LiDAR for auto-labeling, the accuracy of depth estimation directly affects label precision. Particularly, errors increase quadratically with distance, resulting in lower-quality pseudo-labels for distant objects and thus limiting our model's performance at longer ranges. This issue merits further investigation in our future research endeavors.

# G  Licenses

Table 13: Licenses of assets used.

| Asset | License |
| --- | --- |
| Cube R-CNN [3] | CC-BY-NC 4.0 |
| Grounding DINO [28] | Apache License 2.0 |
| Segment Anything [24] | Apache License 2.0 |
| Grounded-SAM [47] | Apache License 2.0 |
| GPT-4 [1] | MIT License |
| Unidepth [40] | CC-BY-NC 4.0 |
| Metric3D [67] | 2-Clause BSD License |
| KITTI [14] | CC-BY-NC-SA 3.0 DEED |
| nuScenes [5] | CC-BY-NC 4.0 |
| SUN RGB-D [52] | MIT License |
| ARKitScenes [2] | Apple License (Link) |

# H  Limitations

Our proposed OVM3D-Det framework relies on advanced depth estimation models to generate pseudo-LiDAR. Although state-of-the-art depth estimation models have demonstrated strong generalization capabilities, their predictions for distant objects are not yet accurate enough, which may lead to errors when generating pseudo labels for these distant objects. Additionally, the performance of monocular systems may degrade under adverse lighting or weather conditions, as these factors directly affect the quality and clarity of the input images. These limitations underscore the need for future research to enhance the robustness and versatility of open vocabulary monocular 3D detection systems, particularly in challenging and dynamic environments.

# I  Broader Impact

The end results of this research are a novel approach to 3D detection that can be applied to autonomous driving, augmented reality, and other robotics applications. By enabling improved 3D object detection using monocular images, our framework has the potential to enhance the safety and accuracy of this system, contributing to the overall advancement of autonomous technologies. Before deployment, it is crucial to establish appropriate safety thresholds and conduct thorough testing to ensure reliability. While our approach does not explicitly exploit dataset biases, it is subject to the same limitations and potential biases as other machine learning techniques.

