# OpenReview forum: "Training an Open-Vocabulary Monocular 3D Detection Model without 3D Data"
_NeurIPS.cc/2024/Conference — NeurIPS 2024 poster_

### Official Review · Reviewer_2uq9 · 2024-06-23

**Soundness:** 3
**Presentation:** 3
**Contribution:** 3
**Rating:** 5
**Confidence:** 3

**Summary:**

This paper introduces OVM3D-Det, an open-vocabulary monocular 3D object detection framework that utilizes only RGB images for training. It leverages open-vocabulary 2D models and pseudo-LiDAR to automatically label 3D objects in these images. Additionally, it incorporates pseudo-LiDAR erosion and bounding box refinement modules, enhanced by prior knowledge from large language models. Experimental results demonstrate the effectiveness of the proposed method.

**Strengths:**

1. This paper is well-written. It’s innovative to introduce the LLM for assisting in the generation of Pseudo 3D Bounding Boxes.
2. The Box Orientation Estimation and Bounding Box Search each have their unique designs.

**Weaknesses:**

1. Although the experiments indicate improvements with the proposed method, the overall AP remains low, and the final performance heavily depends on the Depth Estimator used. Moreover, the experiments seem to lack performance metrics for Unidepth.

2. The effect of GPT-4 in this method is minimal, as it only provides object scaling. The costs and benefits of using it are not well-delineated.

**Questions:**

1. Does this framework heavily depend on the performance of Unidepth? What is the detection performance of Unidepth? How significant is the impact of missed objects, such as those at a distant range, on the final training performance?

2. What are the costs and benefits of using GPT-4?  Is the improvement from using LLM minimal in the ablation study? The performance is very close to using an average vehicle scaling size in the ablation study?

3. The ablation studies are confusing. What are the improvements of each component compared to the baseline?

**Limitations:**

According to the ablation study, the proposed modules show improvements.

---

> ### Author Rebuttal · Authors · 2024-08-07
>
> We would first like to express our appreciation for your time and insightful comments. Please find our response to your concerns in the following:
>
> ---
>
> 1. **Performance of Unidepth.**
>
> Thanks for your valuable suggestion. We experiment with different architectures of Unidepth, which exhibit different performance. We find that **the more accurate the depth estimation model use, the better the performance of the trained open-vocabulary monocular 3D detection model.** This also indicates that with the development of increasingly powerful depth estimation models, our method has a broad prospect.
>
> | KITTI | Depth Estimation (d1 (higher is better)) | Detection (AP) |
> | --- | --- | --- |
> | Unidepth-ConvNext | 97.2 | 18.2 |
> | Unidepth-Vit | 98.6 | 18.5 |
>
> | SUN RGB-D | Depth Estimation (d1 (higher is better)) | Detection (AP) |
> | --- | --- | --- |
> | Unidepth-ConvNext | 94.8 | 10.3 |
> | Unidepth-Vit | 96.6 | 10.6 |
>
> ---
>
> 2. **Costs and benefits of GPT-4.**
>
> We would like to clarify that since we do not use any annotations in the dataset during training, ***we cannot obtain prior information about objects from the dataset***. Therefore, we propose to utilize a large language model, which has seen many descriptions of object shapes on the web, to provide reasonable shape priors. As shown in Table 3(d), using LLM-generated priors yields comparable results to utilizing dataset statistics. This indicates that ***LLMs indeed provide a reasonable estimation of object shape priors***.
>
> Please note that our access to GPT-4 is not on a sample-wise basis; instead, we only need to access GPT-4 once for the priors of all the categories we interested. ***The cost can be negligible.*** Furthermore, we can utilize open-source large language models, such as LLaMA.
>
> ---
>
> 3. **Missing of distant objects.**
>
> If distant objects are missing, it can lead to confusion during the model training.
>
> However, we find that the self-training approach can continuously refine the pseudo labels, which involves employing the results from the previously well-trained model as pseudo labels in subsequent training processes, thereby enhancing the quality of the initially generated pseudo boxes. As shown in the table below, ***self-training can further enhance the model's performance across various distances.***
>
> | Model | Performance on KITTI (AP) | AP-Near | AP-Middle | AP-Far |
> | --- | --- | --- | --- | --- |
> | Ours | 18.5 | 35.6 | 22.0 | 4.5 |
> | Ours + self-train | 19.9 | 37.5 | 24.6 | 6.1 |
>
> Additionally, we can utilize techniques introduced in previous work [1], zooming in regions in the image that contains small or distance objects to improve the 3D detection performance of these instances.
>
> ---
>
> 4. **Ablation studies.**
>
> We apologize for any confusion. In Table 3, our framework's default setting is indicated in grey, with each sub-table representing the ablation of one component. For example, in Table 3(b), compared to the fixed erosion method, our adaptive erosion component bring an increase of 2.1 AP (16.4 vs 18.5). We will improve the presentation of our ablation studies in the revision.
>
> [1] ZoomNet: Part-Aware Adaptive Zooming Neural Network for 3D Object Detection. Xu et al. AAAI 2020.

---

> > ### Author Response · Authors · 2024-08-12
> >
> > Thank you for your insightful comments. We have carefully considered your concerns and questions, and we have made every effort to address them thoroughly in our rebuttal. Since the discussion phase is halfway through, we would like to know if our responses have adequately addressed your concerns.
> >
> > We would greatly appreciate it if you could reassess the paper in light of our responses, provided they have clarified your concerns. If not, we would be happy to provide further explanations or clarifications on any part of the paper. Your feedback is invaluable to us, and we are committed to addressing all your inquiries comprehensively.
> >
> > Once again, thank you for your time and effort in reviewing our paper and for your valuable feedback.

---

> > > ### Comment · Reviewer_2uq9 · 2024-08-13
> > >
> > > Thanks for your rebuttals. I have read all the contents, and most of my concerns have been addressed.

---

> > > > ### Author Response · Authors · 2024-08-14
> > > >
> > > > Thank you for your valuable feedback and support. We are committed to incorporating the revisions in the final version.

---

### Official Review · Reviewer_Rjjg · 2024-07-02

**Soundness:** 3
**Presentation:** 3
**Contribution:** 3
**Rating:** 5
**Confidence:** 5

**Summary:**

This paper proposes OVM3D-Det, an open-vocabulary monocular 3D object detection framework that trains detectors using only RGB images. It introduces two key designs: adaptive pseudo-LiDAR erosion and bounding box refinement, addressing challenges arising from the absence of LiDAR point clouds.

**Strengths:**

1.	The paper introduces an innovative open-vocabulary monocular 3D object detection framework that utilizes only RGB images for training.
2.	The proposed method demonstrates superior performance compared to existing state-of-the-art approaches in both indoor and outdoor scenarios.

**Weaknesses:**

1.	The KITTI dataset has a limited number of classes (5 classes). It would be beneficial to provide results on the NuScenes dataset (23 classes) to better demonstrate the open-vocabulary performance.
2.	The adaptive pseudo-LiDAR erosion module is not illustrated in sufficient detail. The method section discussing this module is vague; please provide more comprehensive details on its design.
3.	More visualizations of the pseudo-LiDAR erosion module are needed. Currently, the visualization of this module is only included in the overall framework figure. Additional visualizations would better demonstrate its effectiveness.
4.	The related work section on monocular 3D object detection is not up-to-date. It is recommended to cite recent papers accepted by CVPR 2024:
a)	MonoCD: Monocular 3D Object Detection with Complementary Depths, CVPR24
b)	Learning Occupancy for Monocular 3D Object Detection, CVPR24
c)	Weakly Supervised Monocular 3D Detection with a Single-View Image, CVPR24
d)	Decoupled Pseudo-labeling for Semi-Supervised Monocular 3D Object Detection, CVPR24
5.	For Table 1 and Table 2, it would be helpful to include a baseline trained without ground-truth annotations for base classes to provide a better comparison.

**Questions:**

Please refer to the weakness section.

---

> ### Author Rebuttal · Authors · 2024-08-07
>
> We would first like to express our appreciation for your time and insightful comments. Please find our response to your concerns in the following:
>
> ---
>
> 1. **nuScenes result.**
>
> We have conducted experiments on nuScenes, and our method has consistently achieved good results, demonstrating its generalizability. **Please refer to the results in the global rebuttal.**
>
> ---
>
> 2. **Details of adaptive pseudo-LiDAR erosion module.**
>
> Thanks for your valuable suggestion. We will add these details below and **the diagram in the attached PDF** to the Sec 3.1 of the revision:
>
> We propose to employ the image erosion operation to eliminate inaccurate object mask edges,
> thereby obtaining accurate pseudo-LiDAR after unprojection.
> Image erosion is a fundamental operation in digital image processing that involves shrinking or wearing away the boundaries of foreground objects in a binary image.
> $M_i$ is the binary mask of one object we interest, where each pixel $M_i(x,y)$ is either 0 (background) or 1 (foreground).
> Let $B(u,v)$ represent the elements of the structuring element $B$, which is a small binary matrix of size $m \times n$.
> Erosion of the mask $M_i$ by the structuring element $B$ is denoted as $M_i \ominus B$.
> $$
> (M_i \ominus B)(x, y) =
> \begin{cases}
> 1 & \text{if } M_i(x+u, y+v) = 1 \text{ for all } (u,v) \text{ such that } B(u,v) = 1, \\\\
> 0 & \text{otherwise.}
> \end{cases}
> $$
> In our experiment, we employ a 3x3 structuring element $B$ where all elements are set to 1.
> The erosion operation can be applied repeatedly.
> The intensity of erosion can be controled by adjusting the number of erosion iterations.
> We note that aggressive erosion may completely remove small objects like distant pedestrians,
> while mild erosion may leave considerable noise on large objects.
> Therefore, for masks with a smaller area, we use a smaller number of iterations;
> conversely, for masks with a larger area, we use a larger number of iterations, ensuring more accurate pseudo-LiDAR.
>
> ---
>
> 3. **More visualization of pseudo-LiDAR erosion module.**
>
> **Please refer to the more visualizations in the attached PDF**, and we will incorporate them into the camera-ready version.
>
> ---
>
> 4. **Related works.**
>
> Thank you for pointing out. We will make sure add them in the revision.
>
> ---
>
> 5. **Baseline trained without ground-truth for base classes.**
>
> Thanks for your valuable advice. We compare our framework to the open-vocabulary 3D detection method OV-3DET [1]. OV-3DET is trained and tested on point cloud data, but ***it does not require any annotations for training***. To adapt it to the monocular detection scenario, we ***use pseudo-LiDAR as its input***. Furthermore, to mitigate the data distribution shift between training and test for OV-3DET, we ***train the OV-3DET framework using pseudo-LiDAR***.
>
> However, their framework is designed to generate pseudo labels from real point clouds, ***failing to address the inherent noise in pseudo-LiDAR data***, which results in low-quality labels.
>
> In contrast, ***our framework is tailored to the noisy nature of pseudo-LiDAR***, incorporating adaptive pseudo-LiDAR erosion and bounding box refinement techniques, and leveraging prior knowledge from large language models, which allows us to generate pseudo labels more effectively for pseudo-LiDAR. **Please refer to the results in the global rebuttal.**
>
> [1] Open-Vocabulary Point Cloud Object Detection without 3D Annotation. Lu et al. CVPR 2023.

---

> > ### Comment · Reviewer_Rjjg · 2024-08-12
> >
> > Thank you for providing the additional experiments and detailed explanations in your rebuttal. It addressed my concerns effectively. I recommend the authors incorporate all new results in the rebuttal into the final version.

---

> > > ### Author Response · Authors · 2024-08-12
> > >
> > > Thank you very much for your insightful feedback. We are grateful for your recommendation to incorporate the new results into the final version, and we will certainly do so.
> > >
> > > We would greatly appreciate it if you could kindly express a clearer indication of your acceptance intention, as it could significantly impact the outcome of our submission. Thank you again for your time and consideration.

---

### Official Review · Reviewer_UAne · 2024-07-13

**Soundness:** 3
**Presentation:** 3
**Contribution:** 3
**Rating:** 6
**Confidence:** 5

**Summary:**

The paper presents a novel open-vocabulary monocular 3D object detection framework named OVM3D-Det. This approach aims to train 3D object detectors using only RGB images, making it cost-effective and scalable. The framework utilizes pseudo-LiDAR and large language models to generate pseudo 3D labels, enabling training without high-precision LiDAR data. The authors propose adaptive pseudo-LiDAR erosion and bounding box refinement to improve label accuracy. Extensive experimentation demonstrates the superiority of OVM3D-Det over existing baselines in both indoor and outdoor scenarios.

**Strengths:**

- **Innovative Approach**: The paper introduces a novel method for open-vocabulary 3D object detection using only RGB images, which significantly reduces the cost and complexity of data acquisition.
 &nbsp;
- **Effective Techniques**: The adaptive pseudo-LiDAR erosion and bounding box refinement with prior knowledge from large language models are innovative solutions that address the challenges of noisy point clouds and occluded objects.
&nbsp;
- **Comprehensive Experiments**: The authors provide extensive experimental results, demonstrating the effectiveness of their approach across different datasets and scenarios.
&nbsp;
- **Broad Applicability**: The proposed method has broad applications in autonomous driving, robotics, and augmented reality, making it a valuable contribution to the field.

**Weaknesses:**

- **Dependence on Depth Estimation Quality**: The performance of the proposed method heavily relies on the quality of the depth estimation model, which may not always be accurate, especially for distant objects.
 &nbsp;
- **Limited Evaluation on Adverse Conditions**: The paper does not extensively compare the method with other indoor OV-3Det methods [1, 2]
 &nbsp;
- **Complexity of Implementation**: The framework involves several sophisticated components, including depth estimation, pseudo-LiDAR generation, and large language models, which might complicate implementation and reproducibility.
 &nbsp;

[1] Yuheng Lu, Chenfeng Xu, Xiaobao Wei, Xiaodong Xie, Masayoshi Tomizuka, Kurt Keutzer, and Shanghang Zhang. Open-vocabulary point-cloud object detection without 3d annotation. In CVPR, 2023. 1, 3
[2] Yang Cao, Zeng Yihan, Hang Xu, and Dan Xu. Coda: Collaborative novel box discovery and cross-modal alignment for open-vocabulary 3d object detection. In NeurIPS, 2023

**Questions:**

- How does the performance of the proposed method vary with the quality of the depth estimation model? Can the authors provide more insights or quantitative analysis on this aspect?
 &nbsp;
- Have the authors considered comparing the indoor OV-3Det methods[1, 2]?
 &nbsp;
- Can the authors provide more details on the computational requirements and efficiency of the proposed framework? How does it compare with point cloud-based methods in terms of computational costs?
 &nbsp;
- How generalizable is the proposed method to other types of data or domains beyond those tested in the experiments?
 &nbsp;
- Are there any specific challenges or limitations in integrating the proposed method into real-world applications such as autonomous driving systems?
 &nbsp;

[1] Yuheng Lu, Chenfeng Xu, Xiaobao Wei, Xiaodong Xie, Masayoshi Tomizuka, Kurt Keutzer, and Shanghang Zhang. Open-vocabulary point-cloud object detection without 3d annotation. In CVPR, 2023. 1, 3
[2] Yang Cao, Zeng Yihan, Hang Xu, and Dan Xu. Coda: Collaborative novel box discovery and cross-modal alignment for open-vocabulary 3d object detection. In NeurIPS, 2023

**Limitations:**

The authors have provided discussions about the limitations and broader impact.

---

> ### Author Rebuttal · Authors · 2024-08-07
>
> We would first like to express our appreciation for your time and insightful comments. Please find our response to your concerns in the following:
>
> ---
>
> 1. **Dependence on depth estimation quality.**
>
> Indeed, our method relies on accurate depth estimation, similar to most monocular 3D detection methods. To address the issue of potential inaccuracies in estimating the depth of distant objects:
>
> - We can continuously refine the pseudo labels using a self-training approach, which involves employing the results from the previously well-trained model as pseudo labels in subsequent training processes, thereby enhancing the quality of the initially automatically generated pseudo boxes. As shown in the table below, ***self-training can further enhance the model's performance across various distances***.
>
> | Model | Performance on KITTI (AP) | AP-Near | AP-Middle | AP-Far |
> | --- | --- | --- | --- | --- |
> | Ours | 18.5 | 35.6 | 22.0 | 4.5 |
> | Ours + self-train | 19.9 | 37.5 | 24.6 | 6.1 |
> - We can utilize techniques introduced in previous work [1], zooming in regions in the image that contains small or distance objects to improve the 3D detection performance of these instances.
>
> ---
>
> 2. **Performance vary with the quality of the depth estimation.**
>
> Thanks for your insightful comment. We experiment with different architectures of Unidepth, which exhibit different performance. We find that ***the more accurate the depth estimation model use, the better the performance of the trained open-vocabulary monocular 3D detection model.*** This also indicates that with the development of increasingly powerful depth estimation models, our method has a broad prospect.
>
> | KITTI | Depth Estimation (d1 (higher is better)) | Detection (AP) |
> | --- | --- | --- |
> | Unidepth-ConvNext | 97.2 | 18.2 |
> | Unidepth-Vit | 98.6 | 18.5 |
>
> | SUN RGB-D | Depth Estimation (d1 (higher is better)) | Detection (AP) |
> | --- | --- | --- |
> | Unidepth-ConvNext | 94.8 | 10.3 |
> | Unidepth-Vit | 96.6 | 10.6 |
>
> ---
>
> 3. **Evaluation with other OV-3Det methods.**
>
> Thanks for your valuable suggestion. We have provided the comprehensive comparison with OV-3DET [2] **in the global rebuttal above** and will add them in the revision. Considering OV-3DET [2] being trained and tested on point cloud data, to adapt it to the monocular detection scenario, we ***use pseudo-LiDAR as its input***. Furthermore, to mitigate the data distribution shift between training and test for OV-3DET, we ***train the OV-3DET framework using pseudo-LiDAR***.
>
> However, their framework is designed to generate pseudo labels from real point clouds, without taking into account the noisy nature of pseudo-LiDAR, which results in low-quality labels.
>
> In contrast, ***our framework is tailored to the noisy nature of pseudo-LiDAR***, incorporating adaptive pseudo-LiDAR erosion and bounding box refinement techniques, and leveraging prior knowledge from large language models, which allows us to generate pseudo labels more effectively for pseudo-LiDAR.
>
> CoDA [3] also faces the same difficulties as OV-3DET when adapting to pseudo-LiDAR. Additionally, CoDA requires training with annotations from base categories, but our method does not require any annotations, so it is unfair to directly compare with them.
>
> ---
>
> 4. **Generalize to other data.**
>
> We have conducted experiments on additional datasets such as ARKitScenes and nuScenes, demonstrating the effectiveness of our method.
>
> We make no modifications to our framework, ***using exactly the same parameters*** on ARKitScenes and nuScenes as we do on SUN RGB-D and KITTI to generate pseudo labels. Despite ***ARKitScenes and nuScenes having completely different data distributions from SUN RGB-D and KITTI*** (due to different sensor models and parameters, different shooting angles, weather conditions, and geographical locations), our framework is still able to generate effective pseudo labels and achieve good results. **Please refer to the global rebuttal for the numbers.**
>
> ---
>
> 5. **Computational cost.**
>
> Thanks for your suggestion. We compare our computational cost on SUN RGB-D dataset to OV-3DET [2] in the table below. Since we do not infer 3D point clouds but only infer images, the inference latency is significantly reduced.
>
> | Model | OV-3DET | Ours |
> | --- | --- | --- |
> | Generate pseudo labels | ~2.2h with 1 RTX3090 | ~2.5h with 1 RTX3090 |
> | Training | ~34h with 2 A100s | ~12h with 2 A100s |
> | Inference | 0.08s/sample on A100 | 0.03s/img on A100 |
>
> ---
>
> 6. **Challenges in real-world applications.**
>
> When applying in the real world, it is important to consider the differences in hardware, such as camera parameters and distortion issues that can lead to differences in imaging. Calibration of the hardware should be conducted before application.
>
> Additionally, it would be best to fine-tune the depth model to achieve more accurate estimation results.
>
> ---
>
> 7. **Complexity of Implementation.**
>
> We include details of our implementation in the paper and will release the code to make sure the reproducibility.
>
> [1] ZoomNet: Part-Aware Adaptive Zooming Neural Network for 3D Object Detection. Xu et al. AAAI 2020.
>
> [2] Open-Vocabulary Point Cloud Object Detection without 3D Annotation. Lu et al. CVPR 2023.
>
> [3] CoDA: Collaborative Novel Box Discovery and Cross-Modal Alignment for Open-Vocabulary 3D Object Detection. Cao et al. NeurIPS 2023.

---

> > ### Comment · Reviewer_UAne · 2024-08-11
> > **Further reply**
> >
> > Thank the authors for the effort in the rebuttal. I have carefully read all the contents, and most of my concerns have been addressed. However, I have a question: as far as I know, CoDA doesn't strongly depend on ground truth and can also be trained using pseudo labels. Could you please provide the results for that?

---

> > > ### Author Response · Authors · 2024-08-12
> > >
> > > We greatly appreciate your time and valuable suggestions. We have carefully and thoroughly revisited the CoDA [1] paper and noticed that it includes a comparative analysis with OV-3DET [2] in the final part of the experiments. Following its protocol, we conducted the experiment and present the results below. Our method still significantly outperforms CoDA, as we carefully considered the noisy nature of pseudo-LiDAR, leading to the generation of more accurate pseudo labels.
> > >
> > > Thank you once again for your insightful review, which has greatly enhanced the quality and clarity of our paper. We welcome any further discussion and suggestions.
> > >
> > > | Model | Performance on SUN RGB-D (AP) |
> > > | --- | --- |
> > > | Cube R-CNN (fully-supervised) | 15.1 |
> > > | OV-3DET | 7.1 |
> > > | CoDA | 7.7 |
> > > | Ours | 10.6 |
> > >
> > > [1] CoDA: Collaborative Novel Box Discovery and Cross-Modal Alignment for Open-Vocabulary 3D Object Detection. Cao et al. NeurIPS 2023.
> > >
> > > [2] Open-Vocabulary Point Cloud Object Detection without 3D Annotation. Lu et al. CVPR 2023.

---

> > > > ### Comment · Reviewer_UAne · 2024-08-13
> > > > **Final score**
> > > >
> > > > Thanks to the authors for conducting more experiments to answer my questions. Please include all the results in the final version, which will benefit the community and inspire more work. Totally speaking, most of my questions are addressed well. I raise my score to 6. Good luck :)

---

> > > > > ### Author Response · Authors · 2024-08-13
> > > > >
> > > > > Thank you for your valuable feedback and support. We are committed to incorporating the revisions in the next manuscript.

---

### Official Review · Reviewer_DUxk · 2024-07-13

**Soundness:** 2
**Presentation:** 3
**Contribution:** 2
**Rating:** 6
**Confidence:** 5

**Summary:**

This paper presents OVM3D-Det, a framework for generating 3D bounding box pseudo-labels from monocular RGB images using foundational vision models like GroundingSAM and UniDepth. Using these pseudo-labels, authors train an open-vocabulary version of Cube-RCNN. Authors demonstrate that their proposed approach generalizes to novel categories unseen during training.

**Strengths:**

Problem Motivation. Open-vocabulary 3D detection is an important problem of interest for a wide variety of applications. This is a challenging open problem which requires further study.

Simple Approach. Authors propose a simple and straightforward approach for generating 3D bounding box pseduo-labels and training an open-vocabulary 3D detector.

Well Written. The paper is written clearly and the provided figures (particularly Figure 3) clearly explain the proposed approach.

**Weaknesses:**

Limited Evaluation. Authors claim that there are no prior methods that address this problem. However, since the proposed method trusts monocular depth estimates as if they were ground truth LiDAR, one can simply evaluate existing RGB + LiDAR open vocabulary detector [1] using pseudo-lidar. Notably, there has been significant prior work in open-vocabulary 3D detection which should be acknowledged in this work.

"Closed World" Setup. Splitting datasets into base vs. novel splits goes against the setup of open-vocabulary detection. Notably, in 2D open vocabulary detection, methods pre-train on diverse datasets like V3Det, Objects365, ODinW, and report zero-shot results on all classes in COCO. A similar setup here would be to train on diverse 3D datasets and report zero-shot results on all classes in KITTI and SUNRGBD.

Pseudo-Labeler Heuristics. The design of the pseudo-labeler is extremely brittle, particularly because it relies on hand designed heuristics. I would not expect such a pseudo-labeler to work well on COCO (despite CubeRCNN doing a reasonable job).

Small Scale Evaluation. Both SUNRGBD and KITTI are very small 3D detection datasets by modern standards. These datasets alone are not enough to study open-vocabulary detection. I would encourage authors to train on and evaluate with more diverse datasets. Notably, Omni3D, the dataset the accompanies CubeRCNN, already provides large-scale training data. Why were the other datasets in Omni3D not used?

[1] Open-Vocabulary Point Cloud Object Detection without 3D Annotation. Lu et. al. CVPR 2023.

**Questions:**

Limitations of Base and Novel Splits. Splitting individual datasets into base and novel splits no longer makes sense in the era of foundation models. Explicitly trying to avoid data leakage will prevent methods from using the latest and greatest models, artificially hampering progress. The best way to address this issue is by evaluating on a sufficiently large and diverse test set such that even if a small part of the data has already been seen by some model (e.g. like UniDepth), the aggregate performance is still a reliable indicator of overall test performance.

Realistic Benchmarking on Out of Distribution Datasets. Similar to 2D open-vocabulary detectors, it would be interesting to benchmark the performance of this approach on unseen datasets (not just unseen classes). For example, how might the proposed model perform on nuScenes if it was trained on KITTI? Notably, nuScenes has many classes not included in KITTI like construction vehicle and motorcycle.

Dimension Prior vs. Real Prior. Using LLMs to generate 3D shape priors is an interesting idea. Intuitively, LLMs seen many descriptions of object shapes on the web, so it makes sense that they can provide reasonable shape priors. How do the LLM shape priors (e.g. L, W, H) actually compare with the real shape priors learned from dataset statistics?

Why Pseudo-Label 3D Datasets? Although the premise of the paper makes sense, the execution is a bit confusing. In particular, both KITTI and SUNRGBD both already have 3D bounding box annotations. It doesn't make sense to pseudo-label these datasets. Instead, it makes more sense to try to pseudo-label datasets like COCO that cannot currently be used by existing 3D detectors.

**Limitations:**

Yes, authors highlight that their proposed approach is reliant on depth estimation models, and is susceptible to their failure modes.

---

> ### Author Rebuttal · Authors · 2024-08-07
>
> We would first like to express our appreciation for your time and valuable comments. Please find our response to your concerns in the following:
>
> ---
>
> 1. **Evaluation on OV-3DET and larger-scale datasets.**
>
> Thanks for your suggestion. We show the results **in the global rebuttal** and will add them in the final revision.
>
> ---
>
> 2. **Experimental setup.**
>
> Indeed, ideally, 3D open-world detection should be as generalized as 2D open-world detection, *i.e.*, a foundation model can be applied to any class. However, this generalization has not been feasible yet in 3D for the two main reasons:
>
> - ***The volume of data and the number of categories in 3D datasets are far less than those in 2D datasets.***
> - ***There is an enormous domain gap between different 3D datasets due to sensor configurations.***
>
> Especially for monocular 3D detection, the camera parameters can vary across different datasets, leading to poor transferability [1,2]. Therefore, we follow the current paradigm in many recent 3D open-world detection works which train models on a single dataset and test it on this dataset [3,4,5].
>
> We would like to clarify that ***we do not need to split the datasets into base and novel splits to train our model***. We introduce this base/novel setting only to compare our method with the constructed Cube R-CNN+Grounding DINO baseline due to the lack of a suitable baseline before. In fact, we do not need any ground-truth annotation, as shown in the last line in Tables 1 and 2 of our paper. Therefore, our framework is fully capable of applying and achieving open-world 3D detection when the data are sufficient.
>
> ---
>
> 3. **Why pseudo-label 3D datasets?**
>
> Due to the scale ambiguity inherent in monocular depth estimation [6,7], it is always better to know the camera's focal length to perform accurate monocular depth estimation [8,9]. Therefore, we select datasets with camera focal lengths for our experiments. As for the COCO dataset, we do not know its camera focal lengths.
>
> It is worth noting that our method can generalize beyond 3D datasets. It is to demonstrate our approach for us to conduct experiments on 3D data. In fact, our method can leverage any images along with their camera focal lengths for training. Data acquisition does not necessitate the use of costly LiDAR or 3D scanners.
>
> Additionally, we can use various video data sources like YouTube and the newly released SAM 2 to obtain camera focal lengths through SLAM (Simultaneous Localization and Mapping) to generate pseudo labels and thus facilitate our framework to train 3D open-world detectors on video data.
>
> ---
>
> 4. **Pseudo-labeler heuristics.**
>
> Thanks for your insightful comment. In our framework, the erosion and PCA to determine orientation are both robust. When searching for the bounding box, we assess whether the box falls within the reasonable range of prior knowledge with two thresholds $\tau_1$ and $\tau_2$. Table 3(f)(g) in the paper shows the selection of this threshold is robust and reasonable.
>
> To further validate, we conduct experiments on additional datasets. We make no modifications to our framework, ***using exactly the same parameters*** on ARKitScenes and nuScenes as we do on SUN RGB-D and KITTI to generate pseudo labels. Despite ***ARKitScenes and nuScenes having completely different data distributions from SUN RGB-D and KITTI*** (due to different sensor models and parameters, different shooting angles, weather conditions, and geographical locations), our framework is still able to generate effective pseudo labels and achieve good results. **Please refer to the results in the global rebuttal.**
>
> Additionally, we can refine the model using a self-training approach, which involves using the results from the previously well-trained model as pseudo labels in the subsequent training process. As shown in the table below, ***self-training can further enhance the quality of the initially generated pseudo boxes***.
>
> | Model | Performance on KITTI (AP) |
> | --- | --- |
> | Ours | 18.5 |
> | Ours + self-train | 19.9 |
>
> ---
>
> 5. **LLM-generated priors vs. real priors.**
>
> Thanks for your suggestion. **In Tables 1 and 2 of the attached PDF,** we provide a detailed comparison of shape priors on KITTI and SUN RGB-D. Since we only utilize the shape priors to filter out pseudo boxes that may be unreasonable due to noise or occlusion and to refine them, as long as the shape priors are within a reasonable range, they are sufficient for our purposes.
>
> ---
>
> 6. **Out of Distribution Dataset.**
>
> In the following table, we present the evaluation on nuScenes using the model trained on KITTI. Unsurprisingly, the results are inferior to those obtained from direct training. Many previous studies indicate that there is a significant domain gap between different 3D datasets. Especially for monocular 3D detection, the camera parameters vary across different datasets, leading to poor transferability [6,7].
>
> | Model | Performance on nuScenes (AP) |
> | --- | --- |
> | Ours (trained on nuScenes) | 13.9 |
> | Ours (trained on KITTI) | 3.2 |
>
> [1] MonoUNI: A Unified Vehicle and Infrastructure-side Monocular 3D Object Detection Network with Sufficient Depth Clues. Jia et al. NeurIPS 2023.
>
> [2] FS-Depth: Focal-and-Scale Depth Estimation from a Single Image in Unseen Indoor Scene. Wei et al. TCSVT 2024.
>
> [3] Open-Vocabulary Point Cloud Object Detection without 3D Annotation. Lu et. al. CVPR 2023.
>
> [4] PLA: Language-Driven Open-Vocabulary 3D Scene Understanding. Ding et al. CVPR 2023.
>
> [5] OpenScene: 3D Scene Understanding with Open Vocabularies. Peng et al. CVPR 2023.
>
> [6] Is Pseudo-Lidar needed for Monocular 3D Object detection? Park et al. ICCV 2021.
>
> [7] Pseudo-LiDAR++: Accurate Depth for 3D Object Detection in Autonomous Driving. You et al. ICLR 2020.
>
> [8] Towards Zero-Shot Scale-Aware Monocular Depth Estimation. Guizilini et al. ICCV 2023.
>
> [9] Metric3D: Towards Zero-shot Metric 3D Prediction from A Single Image. Yin et al. ICCV 2023.

---

> > ### Comment · Reviewer_DUxk · 2024-08-08
> >
> > Authors have sufficiently addressed my questions. I recommend accepting this paper.

---

> > > ### Author Response · Authors · 2024-08-09
> > >
> > > Thanks for your positive feedback! We are glad that we have addressed your concerns.

---

> > > ### Author Response · Authors · 2024-08-12
> > >
> > > We would like to express our gratitude once again for your time and valuable feedback. In our last communication, you mentioned the possibility of improving the rating for our paper. Is there any additional issue that requires our attention? We welcome any further discussion and suggestions.

---

### Author Rebuttal · Authors · 2024-08-07

We thank all reviewers **[R1,DUxk], [R2,UAne], [R3,Rjjg], [R4,2uq9]** for their constructive comments and helpful feedback.

All reviewers agree on the efficacy of our method, including its broad applicability (R1, R2), innovative design (R2, R3, R4), and clarity of writing (R1, R4). They also highlight the extensive experimental results that demonstrate the effectiveness of our approach (R2, R3), and the significant reduction in cost and complexity associated with data acquisition (R2).

We have carefully considered the reviewers' comments and provided additional clarification to address each concern. Here, we offer general responses to all reviewers on two key issues.

---

1. **Open-vocabulary 3D object detection baselines.**

Reviewers [R1, R2, R3] have requested that we ***compare our method with existing state-of-the-art open-vocabulary 3D detection approaches***. R1 suggests that we can simply evaluate existing open vocabulary detector OV-3DET[1] that is trained and tested on point cloud data using pseudo-LiDAR as input. We would like to thank R1 very much for pointing out this straightforward baseline.

Specifically, we only ***replace the point cloud input of OV-3DET with the pseudo-LiDAR*** which is unprojected from the RGB images and estimated depth maps. Other parts of the model in OV-3DET are kept untouched for fair comparison.

As shown in the table below, directly changing the input data format leads to poor performance with only 3.4 AP. This phenomenon is also observed and discussed in previous works [2,3,4], which we attribute to the distribution shift between real point cloud and pseudo-LiDAR.

***To mitigate this data distribution shift, we train the OV-3DET framework using pseudo-LiDAR.*** It improves from 3.4 to 7.1 AP, marking a 2x improvement compared to the original model. Nonetheless, the 10.6 AP of our method still demonstrates our superiority, since ***our approach focuses on adapting to the noisy nature of pseudo-LiDAR***, whereas previous open-vocabulary 3D detectors are mainly designed for real point clouds to generate pseudo labels.

The key designs that distinguish our method from other baselines are the adaptive pseudo-LiDAR erosion and bounding box refinement techniques, incorporating the prior knowledge of language models. Our method can generate pseudo labels more effectively for pseudo-LiDAR than other baselines, as reported in Table 3 (a)(b)(d) of the paper.

| Model | Performance on SUN RGB-D (AP) |
| --- | --- |
| Cube R-CNN (fully-supervised) | 15.1 |
| OV-3DET (original, trained with point cloud) | 3.4 |
| OV-3DET (trained with pseudo-LiDAR) | 7.1 |
| Ours | 10.6 |

For outdoor 3D detection, we also try to establish another baseline since OV-3DET [1] only includes results on indoor data. We attempt to train OV-3DET on the KITTI dataset with pseudo-LiDAR. As shown in the table below, simply training on pseudo-LiDAR points causes a rather weak performance of only 1.3 AP. This result reflects the nature of outdoor data that the ***vast range of spatial scale could lead to intolerable pseudo-label errors from tiny noise.*** Therefore, it is challenging for the baselines to generate reasonable and reliable pseudo boxes, depicted in Fig. 2 of the attached PDF, where the raw pseudo-LiDAR of inaccurate object edges have a very long "tail."

| Model | Performance on KITTI (AP) |
| --- | --- |
| Cube R-CNN (fully supervised) | 31.4 |
| OV-3DET (trained with pseudo-LiDAR) | 1.3 |
| Ours | 18.5 |

---

2. **Results on additional datasets.**

We provide additional results on ARKitScenes and nuScenes to further validate the generalization capability of our proposed framework. Without any modifications to our framework, we ***use exactly the same parameters*** on ARKitScenes and nuScenes as  SUN RGB-D and KITTI to generate pseudo labels.

Although ***ARKitScenes and nuScenes possess completely different data distributions from SUN RGB-D and KITTI*** , including different sensor models and parameters, different shooting angles, weather conditions, and geographical locations, our framework is still able to generate effective pseudo labels and achieve good results, as shown in the tables below.

| Model | Performance on ARKitScenes (AP) |
| --- | --- |
| Cube R-CNN (fully supervised) | 38.5 |
| OV-3DET (trained with pseudo-LiDAR) | 13.5 |
| Ours | 21.2 |

| Model | Performance on nuScenes (AP) |
| --- | --- |
| Cube R-CNN (fully supervised) | 28.8 |
| OV-3DET (trained with pseudo-LiDAR) | 1.1 |
| Ours | 14.1 |

---

**For detailed responses to individual reviewer comments, please refer to our separate responses to each reviewer.**

Lastly, we would like to thank the reviewers for their time and we are welcome for any further discussion.

[1] Open-Vocabulary Point Cloud Object Detection without 3D Annotation. Lu et al. CVPR 2023.

[2] Is Pseudo-Lidar needed for Monocular 3D Object detection? Park et al. ICCV 2021.

[3] Monocular 3D Object Detection via Feature Domain Adaptation. Ye et al. ECCV 2020.

[4] Monocular 3D Object Detection with Pseudo-LiDAR Point Cloud. Weng et al. ICCVW 2019.

---

### Decision · Program_Chairs · 2024-09-25

**Decision:**

Accept (poster)

**Comment:**

Post-rebuttal, all four reviewers are in favor of acceptance. The AC has examined the paper, the reviews, rebuttal, and discussion and concurs. Since the rebuttal was critical for the paper's acceptance, the AC urges the authors to carefully incorporate all information from the rebuttal into the final version of the paper.